# Zero Pixel Directional Boundary by Vector Transform

**Edoardo Mello Rella**[1], **Ajad Chhatkuli**[1], **Yun Liu**[1], **Ender Konukoglu**[1] **& Luc Van Gool**[1,2]
[1] Computer Vision Lab, ETH Zurich, Switzerland    [2] VISICS, ESAT/PSI, KU Leuven, Belgium

## Abstract

Boundaries are among the primary visual cues used by human and computer vision systems. One of the key problems in boundary detection is the label representation, which typically leads to class imbalance and, as a consequence, to thick boundaries that require non-differential post-processing steps to be thinned. In this paper, we re-interpret boundaries as 1-D surfaces and formulate a one-to-one vector transform function that allows for training of boundary prediction completely avoiding the class imbalance issue. Specifically, we define the boundary representation at any point as the unit vector pointing to the closest boundary surface. Our problem formulation leads to the estimation of direction as well as richer contextual information of the boundary, and, if desired, the availability of zero-pixel thin boundaries also at training time. Our method uses no hyper-parameter in the training loss and a fixed stable hyper-parameter at inference. We provide theoretical justification/discussions of the vector transform representation. We evaluate the proposed loss method using a standard architecture and show the excellent performance over other losses and representations on several datasets.

## 1 Introduction

Boundaries are important interpretable visual cues that can describe both the low-level image characteristics as well as high-level semantics in an image. Human vision uses occluding contours and boundaries to interpret unseen or seen objects and classes. In several vision tasks, they are exploited as priors (Zhu et al., 2020; Kim et al., 2021; Hatamizadeh et al., 2019; Revaud et al., 2015; Cashman & Fitzgibbon, 2012). Some key works on contours (Cootes et al., 2001; Matthews & Baker, 2004; Kass et al., 1988) have greatly impacted early research in computer vision. Although the advent of end-to-end deep learning has somewhat shifted the focus away from interpretable visual cues, boundary discovery still remains important in computer vision tasks.

Supervised deep learning has greatly transformed problems such as object detection and segmentation (Redmon et al., 2016; Chen et al., 2017; Cheng et al., 2020) by redefining the problem (Kirillov et al., 2019), using high-quality datasets (Cordts et al., 2016; Neuhold et al., 2017) and better network architectures (Cheng et al., 2020; 2021; Wang et al., 2021b). Boundary detection, however, has seen a rather modest share of such progress. Although, modern deeply learned methods (Xie & Tu, 2015; Liu et al., 2017; Maninis et al., 2017) provide better accuracy and the possibility to learn only the high-level boundaries, a particularly elusive goal in learned boundary detection has been the so-called crisp boundaries (Isola et al., 2014; Wang et al., 2018; Deng et al., 2018). The formulation of boundary detection as a binary segmentation task naturally introduces class imbalance, which makes detecting pixel thin boundaries extremely difficult. Arguably a majority of recent methods in boundary detection are proposed in order to tackle the same issue. Many methods address the lack of 'crispness' by fusing high-resolution features with the middle- and high-level features (Xie & Tu, 2015; Liu et al., 2017). Such a strategy has been successful in other dense prediction tasks (Ronneberger et al., 2015) as well. Others propose different loss functions (Kokkinos, 2016; Deng et al., 2018; Kervadec et al., 2019) to address class imbalance.

Despite the improvements, we identify two issues regarding crisp boundary detection. The first is that the evaluation protocol (Martin et al., 2004) does not necessarily encourage crisp detection, as the quantification is done after Non-Maximal Suppression (NMS). Such an evaluation may be misleading when the network outputs need to be used at training time for other high-level tasks,

e.g., segmentation (Kim et al., 2021). Second, the current losses (Kokkinos, 2016; Xie & Tu, 2015; Ma et al., 2021) push for edges as crisp as the ground-truth rather than as crisp as possible. This is particularly harmful since many boundary detection datasets (Arbelaez et al., 2010; Silberman et al., 2012) contain ambiguous boundaries or inconsistently thick boundaries.

In this paper, we take a different perspective on boundary detection. Boundaries are formed where visual features change, popularly referred to as the differential representation (Boykov et al., 2006; Kervadec et al., 2019). Such an assumption treats the boundaries as a set of 1-D surfaces embedded in a continuous 2D space, implying they do not have any thickness. Many previous energy-minimization based approaches (Chan & Zhu, 2005; Chan & Vese, 2001; Paragios et al., 2002; Boykov et al., 2006; Ma & Manjunath, 2000) and a few current methods (Kervadec et al., 2019) tackle boundaries in a similar way. Level-set methods (Chan & Zhu, 2005; Boykov et al., 2006; Ma & Manjunath, 2000) consider boundaries as the level-set of a continuous function of the image. Specifically, (Ma & Manjunath, 2000) defines the energy function related to the distance and direction of the boundary at each pixel and extracts the directional normals at the boundary using such an energy. Inspired by such works and also recent works on 3D implicit functions (Tancik et al., 2020; Sitzmann et al., 2020; Mildenhall et al., 2020), we represent boundaries via a field of unit vectors defined at each pixel, pointing towards the closest boundary surface. The proposed vector field representation naturally solves class imbalance. In distance transforms, vector fields are considered incomplete euclidean transforms (Osher & Sethian, 1988), equal to the Jacobian of the signed distance field. The vector field we use is in fact the Jacobian of the positive distance field. In contrast to distance fields, it provides high sensitivity at the boundaries and is easily localizable. We demonstrate the equivalence of the normal field to the surface contour representation using the level set of the normal field's divergence, providing infinitely sharp boundaries. Owing to the zero-thickness, we refer to our result as the zero-pixel boundary. Our method is virtually hyper-parameter free at training and test time, and can provide zero-pixel thick boundaries at training time.

In order to evaluate the boundaries using the surface interpretation, we also advocate the use of surface distances including the average symmetric surface distance (assd) metric that is less prone to class imbalance and variable boundary thickness in the ground-truth (Kervadec et al., 2019). Such metrics are very popular in biomedical image segmentation (Yeghiazaryan & Voiculescu, 2018). We show significant improvements in all metrics using our boundary representation when compared to the various combinations of Dice (Dice, 1945) and cross-entropy losses in several large datasets.

## 2 RELATED WORK

There is a rich history of boundary detection in computer vision. Previous work on boundaries showed a diversity of definitions and approaches. We differentiate them based on two different interpretations of boundaries: i.e., *i)* a boundary is a separation between two or more image regions with different visual features and *ii)* a boundary is a thin group of pixels belonging to a specific class. It should be noted that most modern methods fall under the second category.

Perhaps the most notable representatives of the first strand are the energy-based segmentation methods (Chan & Vese, 2001; Comaniciu & Meer, 2002; Boykov et al., 2006; Grady, 2006; Ma & Manjunath, 2000). These methods relied on hand-crafted features and various optimization strategies to compute the low-level boundaries. In particular, Ma & Manjunath (2000) compute the normal vectors of the low-level boundaries from an energy function, without looking into an equivalent learnable representation. Graph-based segmentation methods (Shi & Malik, 2000; Felzenszwalb & Huttenlocher, 2004; Cheng et al., 2016) construct a graph from the image and cut the graph to obtain non-overlapping regions whose boundaries are viewed as image boundaries. A few deep learning methods followed a similar approach (Wang et al., 2021a; Kervadec et al., 2019). Despite the advantage of the definition, current representations in this category are hard to adapt to generic boundaries and a compact multi-boundary representation with good performance remains lacking.

A larger corpus of work utilizes pixel-wise image features to decide whether pixels belong to a 'boundary' class. They form our category *ii)* methods. Early methods utilize various filter operators to detect discontinuities in image intensities or colors (Canny, 1986; Sobel, 1972). Learning based methods substantially boost the edge detection performance by classifying handcrafted features (Konishi et al., 2003; Martin et al., 2004; Arbelaez et al., 2010; Dollár & Zitnick, 2013; Hallman & Fowlkes, 2015). Modern deep neural network (DNN) methods have further improved

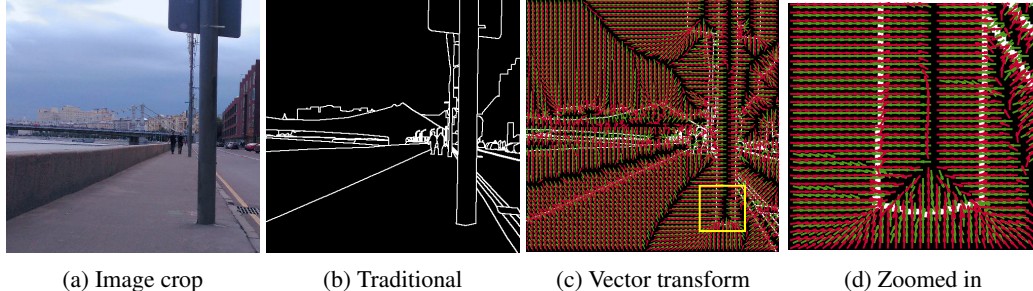

|           |             |                   |             |
|:---------:|:-----------:|:-----------------:|:-----------:|
| (a) Image crop | (b) Traditional | (c) Vector transform | (d) Zoomed in |

Figure 1: **Boundary representations**. We contrast the conventional binary boundary representation with our representation. From left to right, we show (a) top-left crop of an image in a test set, (b) the standard binary representation of ground-truth boundary, (c) the vector transform plot of the prediction *(in red)* overlaid on the ground-truth representation *(in green)* and the conventional binary representation for clarity. Finally (d) shows the zoomed in view of (c) on the yellow rectangle.

this field by learning powerful feature representations, particularly high-level semantic information (Shen et al., 2015; Bertasius et al., 2015; Xie & Tu, 2015; Liu et al., 2017; Wang et al., 2018; Deng et al., 2018; He et al., 2019). Yang et al. (2016) leveraged the powerful deep features to detect only object boundaries. Others try to simultaneously detect edges and predict the semantic class of each edge point, so-called semantic edge detection (Hariharan et al., 2011; Yu et al., 2017; Liu et al., 2018; Yu et al., 2018).

On the other hand, classifying pixels as boundary class introduces class imbalance during training. A common counter-strategy is to use a weighted cross-entropy loss giving the non-boundary pixels a small weight and the boundary class a large weight (Xie & Tu, 2015; Liu et al., 2017; He et al., 2019). Yet, despite an improvement over regular cross-entropy, it does not solve the problem. To thin the boundaries, Non-Maximal Suppression (NMS) is usually adopted. Such methods may be harmful when directly integrated with higher-level tasks such as segmentation (Kim et al., 2021). The Dice loss (Dice, 1945) was thus advocated to generate crisp boundaries before NMS (Deng et al., 2018), but it still produces several pixel thick boundaries and suffers more from missed predictions. Variations of the Dice loss (Shit et al., 2021) have been proposed to counter the missed detections. However, the right approach still depends on the downstream tasks (Zhu et al., 2020; Kim et al., 2021; Hatamizadeh et al., 2019; Shit et al., 2021) and in either case a careful selection of training as well as testing hyper-parameters is required. We provide an alternative approach, motivated by the class imbalance, while having no sensitive hyper-parameter.

## 3 BOUNDARY TRANSFORM AND REPRESENTATION

In this section, we first discuss the surface representation of boundaries as a normal vector field transform and prove its relevant properties. Our boundary representation is inspired by recent work on implicit neural 3D surface representations (Park et al., 2019; Mescheder et al., 2019), energy-based methods on edge detection (Ma & Manjunath, 2000; Boykov et al., 2006) and distance transforms (Osher & Sethian, 1988). In 3D surface representations, a Signed Distance Function (SDF) (Osher & Sethian, 1988) or occupancy map (Mescheder et al., 2019) is used as representation. We instead propose a unit vector field from every point to the closest boundary. This choice is motivated by the high sensitivity and richer boundary context provided by the unit vector field, as shown in our experimental results in § 5. Fig. 1 shows our vector field representation of the ground-truth, with predictions using a standard quiver plot on a sub-sampled set of points.

We assume a continuous boundary image domain $\Omega \subset \mathbb{R}^2$ with the set of boundary points $\{x'\} = \Pi \subset \Omega$. Each point is denoted as a 2-vector $x = (x, y) \in \mathbb{R}^2$. In order to encode boundary properties on the whole image, we compute a signed $x$ and $y$ distance field separately and finally encode only the direction. The result is a unit vector field that represents the boundary. We can express our

boundary representation by the following transform for any point $x \in \Omega$:

$$f(x) = -(x - \arg\min_{x' \in \Pi} d(x, x'))$$

$$v(x) = \frac{f(x)}{\|f(x)\|_2}, \quad \text{if } \|f(x)\|_2 \neq 0, \text{ otherwise } n, \quad n = \lim_{f_x \to 0^+} \frac{f(x)}{\|f(x)\|_2}. \tag{1}$$

Equation (1) defines the transform as a function $v(x) : \Omega \to \mathbb{R}^2$ going from the boundary $\Pi$ to a field representation. Here, $d$ is the distance operator and $f_x$ is the $x$ component of the field $f$. Note that we choose the field vector arbitrarily among the two possible values at the boundary by approaching towards the boundary from the positive $f_x$ value.

We note the following properties of the vector field $v$.

**Property 3.1** *The vector field $v(x)$ is equal to the unit normal field at the boundary.*

**Proof** This is a well known result (Osher & Fedkiw, 2003) and can be proved easily (see equation (2.4) in the reference). The fact that we forcefully choose one normal over its negative directional normal at the boundary points does not affect the statement.

**Property 3.2** *Given a vector field representation $v(x)$ of a boundary, one can obtain the binary boundary representation by considering the following transform:*

$$g(x) = \mathrm{div}\, v(x). \tag{2}$$

*The original boundary set $\Pi$ can then be found by taking the zero level set of $g(x) + 2$, i.e.,*

$$\Pi = L_0(g + 2). \tag{3}$$

**Proof** In the infinitesimal neighborhood of the boundary points, using property 3.1, the vector field is normal to the boundary, provided that good approximate normals can be obtained from equation (1). As the neighborhood size approaches zero, the tangential vector components approach zero around a point for a continuous boundary segment. Thus, around such an infinitesimal neighborhood, the normal fields pointing in opposite direction will subtract perfectly, creating a divergence flow of -2 and around 0 or positive away from boundaries.

Strictly speaking the result holds only for piece-wise smooth surfaces (Osher & Fedkiw, 2003), with lower than -2 divergence possible at discontinuous surface points.

**Property 3.3** *The relation is one-to-one between the binary boundary representation and the proposed vector field representation in a continuous domain.*

**Proof** This property is the result of equation (1), for the forward transform and equation (3) for the inverse transform, providing a one-to-one relation.

Note that the vector field transform as defined in equation (1) has to correct for two different kinds of indeterminate states. The first is on the boundary, that is solved by using the right hand limit so that one of the two opposite directions is chosen consistently. The second is when the infimum operation in equation (1) produces two or more closest points, corrected by choosing any one of the points for the infimum. The vector fields around such points flip directions creating a positive divergence as shown in Fig. 2. More discussions are provided in §5 about the latter, which are in fact helpful for deciding superpixel centers.

The above properties and their proofs are crucial for the validity of the proposed boundary representation and also to go from one representation to another for inference and visualization.

**Vector Transform and the Distance Transform.** In essence, the normalized vector field proposed in equation (1) is another representation of the distance transform. Let $\phi(x) \in \mathbb{R}^+$ define the distance transform, then the vector field $v(x)$ in equation (1) can be obtained by the following partial derivatives (Osher & Sethian, 1988; Osher & Fedkiw, 2003):

$$v(x) = -\nabla \phi(x). \tag{4}$$

One can optimize a given network by minimizing the loss on the distance transform (DT) or SDF (Dapogny & Frey, 2012; Caliva et al., 2019; Park et al., 2019) instead of using the loss on the normalized vector field. Compared to the binary mask, the Vector transform (VT), DT and SDF have an added advantage that they are sensitive to small topological changes. SDF on the other hand, does not support overlapping and open surfaces and is not easily adaptable to the image boundary problem. However, there are several reasons which make DT unsuitable for learning boundaries. During training, when the distance field is close to 0 i.e., around the boundaries, any loss on the DT or SDF loses importance. Apart from the convergence problems of DT with gradient descent Osher & Fedkiw (2003), DT is also hard to localize by thresholding under noise compared to the SDF and VT. In SDF, localizing the surface amounts to finding the zero crossings and in VT, the divergence measure in equation (2) provides an extremely sharp contrast making equation (3) trivial to solve. These differences in the thresholding problem can be seen in Fig. 5 in Appendix B and the experimental results. Additionally, despite reducing the class imbalance compared to binary boundary prediction, DT has an implicit bias to the weighted average of its typical range. On the other hand, a normalized vector field from VT equation (1) is sensitive to the topology similar to a distance field while also being localizable and sensitive at the boundaries, as shown in Fig. 1.

## 4 Boundary Detection with the Vector Field

In this section we provide the details for the construction of the boundary detection method using the representation proposed in §3.

### 4.1 Network Architecture

Most convolutional architectures (Liu et al., 2017; Xie & Tu, 2015) for boundary detection take advantage of both the low-level high resolution features and the deep high-level features, using several fusion strategies. We choose a similar network architecture HR-Net (Wang et al., 2020) which was proposed for segmentation and object detection tasks in high resolution images. We further enrich the high resolution representation provided by HR-Net with an additional skip connection at $\times 2$ downsampling level from the encoder to the decoder. This helps retrieve very high resolution details that are necessary in the boundary prediction task. The output of the HR-Net is first bilinearly upsampled and fused with the skip connection. Inspired by Deng et al. (2018), the high resolution skip connection goes through a ResNeXT module (Xie et al., 2017) before being fused with the decoder signal with a pixel-wise fully connected layer. Finally, the output goes through a convolutional layer and is further upsampled to formulate a prediction at the same resolution as the input. For more details refer to Appendix §F. This architecture allows us to test our method as well as traditional, pixel-wise classification methods, without giving an unfair advantage to one over the other. Fig. 2 shows the simple setup of our complete pipeline with training and inference.

The output of our network is a two channel image which contains, respectively, the x-component $\hat{v}_x$ and the y-component $\hat{v}_y$ of the field prediction corresponding to equation (1). We denote the transform of equation (1) as Vector Transform (VT), upon which we build our method. Although it is possible to use a single angle prediction and representation as in $\theta \in [-\pi, \pi]$, we choose the $x$ and $y$ component representation because it avoids discontinuities in its value as the vector changes. To constrain the network to output predictions in the $[-1, 1]$ range typical of the sine and cosine functions, a hyperbolic tangent activation ($\tanh$) is applied to the output of the network.

**Training loss.** To train the $x$ and $y$ components of VT prediction, the mean-squared error (MSE) loss is used, which can be formulated as

$$\ell_{VT} = \left\| v^{gt} - \hat{v} \right\|_2^2 \tag{5}$$

Unlike many other methods, there exists no hyper-parameters on the loss formulation.

### 4.2 Inference

At inference time, we predict the boundary VT densely on the image. It is entirely possible that for some downstream tasks, such a prediction may already provide the required endpoint. However, to obtain the surface representation of the boundary, we need to use the properties listed in §3.

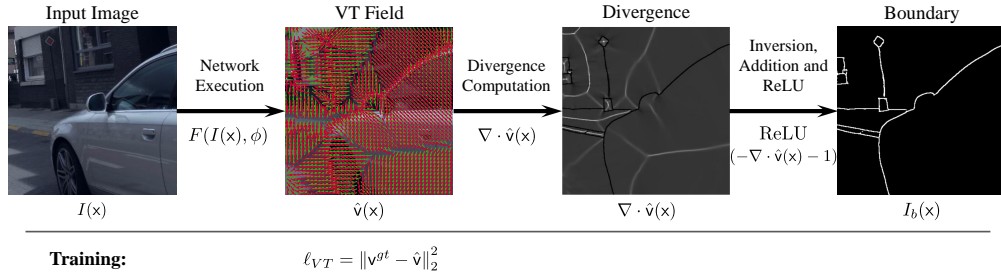

Figure 2: **Training and Inference overview**. The predicted field $\hat{v}(x)$ is convolved with pre-selected filters to obtain the divergence at pixel boundaries for inference. For visualization, we show the divergence and the predicted boundary in pixel locations without using the support image.

In particular, property 3.2 provides a differential means to obtain a zero-thickness boundary on a continuous domain. In the discrete space, the same principle can be used to derive the boundary as the inter-pixel points, which we refer to as the zero-pixel boundary. At this stage, at least two different approaches can be used to obtain the inter-pixel separation. One can be formulated by using $2 \times 1$ or $2 \times 2$ kernels to extract the divergence, which provides the necessary derivatives at the pixel boundaries. We describe in detail a second and simpler approach which uses a support image $\tilde{I}$ of twice the resolution, to extract boundaries as a single pixel detection. We define an operator $Z(I) = \tilde{I}$, which takes in any image $I$ and provides the image $\tilde{I}$ of twice the resolution according to the following rule.

$$Z(I) = \tilde{I}, \quad \begin{cases} \tilde{I}(x,y) = I\left(\frac{x}{2}, \frac{y}{2}\right) & \text{if } (x \bmod 2 = 0 \text{ and } y \bmod 2 = 0), \\ \tilde{I}(x,y) = 0 & \text{otherwise.} \end{cases} \tag{6}$$

Equation (6) is a simple copy and concatenate operation. Using the operator $Z$ on the predicted VT field $\hat{v}$, we obtain $\tilde{v} = Z(\hat{v})$. We then compute the divergence using standard Sobel filters as:

$$\nabla \cdot \tilde{v}(x) = \frac{\partial \tilde{v}(x,y)}{\partial x} + \frac{\partial \tilde{v}(x,y)}{\partial y}. \tag{7}$$

The image with boundaries $I_b$ can then be obtained inverting the divergence of the VT prediction image in equation (7), subtracting 1 from it and applying ReLU activation (Nair & Hinton, 2010):

$$I_b = \text{ReLU}\left(-(\nabla \cdot \tilde{v}(x) + 1)\right). \tag{8}$$

The resulting image will have non-zero values only on the predicted boundary surface, with boundary pixels having values around 1. In practice we obtain small deviations of about 0.1 around the expected value of 1 due to imperfections in the field predictions. In the support image structure, it can be observed that only pixels not belonging to the original image can be classified as boundaries as they are the only ones for which the divergence can have a value different from zero. Note that the above divergence definition using the support image is only required for zero-thickness boundary, we may prefer Fig. 2 for a standard inference process.

In order to evaluate the zero-pixel boundary with traditional metrics, and visualize, boundary pixels need to be represented on the original image. We do so, by copying the values of boundary pixels in the support image to the corresponding neighboring pixels belonging to the original image and averaging the values if necessary; all the other values are set to zero. This leads to two pixel boundaries with the added property of being differentiable, which may be particularly useful when integrating the boundary detection task in a larger deep network. When evaluating the method with the surface distance metrics - which are discussed in §4.3 - the resulting image is thresholded to obtain a binary boundary image. More specifically, all the positive pixels are considered to belong to a boundary and this remains fixed throughout the experiments.

An important aspect of VT is that it provides directional information on each pixel, including those pixels which are associated to the boundaries. Directional boundaries have been previously explored with applications and advantages (Maninis et al., 2017). We provide two key examples. In the first application, we detect boundary pixels which are proposal candidates for straight lines in a differentiable way. In the second, we detect superpixels by grouping pixels inside a convex boundary. These related tasks are discussed more in depth with some qualitative results in the appendices.

### 4.3 METRICS

The standard measures for boundary evaluation (Martin et al., 2004) are fixed contour threshold (ODS) and per-image best threshold (OIS). With these approaches it is possible to compute the recall (R), the precision (P) and the F score. These are the metrics traditionally applied using the approach proposed by Martin et al. (2004) to determine the true positives and false negatives. It proceeds by creating a one-to-one correspondence between the ground-truth boundary pixels and the predicted boundary pixels. In this context, each pixel without a correspondence within a fixed threshold distance is considered either a false positive or a false negative. This approach, however, suffers from a few drawbacks:

- It is extremely sensitive in differences of thickness between the prediction and the ground-truth, as the one-to-one correspondence will not be satisfied resulting in a large number of false positives or false negatives without regard for the actual boundary quality.
- ODS and OIS optimize the threshold hyper-parameter on the test set. This may lead to an unfair evaluation of the quality of the detected boundaries.
- As it is based on pixel correspondences, it cannot be directly applied to zero pixel thin boundaries.

To overcome these drawbacks and have a metric that can be applied to the zero pixel surfaces as well, we propose to use the average surface distances (asd), more favored in the medical imaging community (Yeghiazaryan & Voiculescu, 2018). For every boundary pixel or segment in the prediction, it is the distance to the closest ground truth boundary surface point - either a pixel or a segment. The same can be done starting from the ground truth pixels or segments. The distance from the prediction to the ground truth boundary ($asd_P$) is representative for the quality of the prediction - a precision-like term - while the distance from the ground truth to the prediction ($asd_R$), like recall, shows how effectively all the boundaries are detected. The two scores can be averaged to obtain a single metric, the average symmetric surface distance ($assd$). Given that no one-to-one correspondences between pixels are computed, $asd_P$, $asd_R$ and $assd$ are less sensitive to the boundary thickness and more influenced by the overall quality of the prediction, compared to the previously defined metrics. However, one drawback of the surface distance metrics is that it has high sensitivity to isolated false detections or unmatched ground-truth boundary pixels.

## 5 EXPERIMENTS

We compare the proposed method on Cityscapes (Cordts et al., 2016), Mapillary Vistas (Neuhold et al., 2017) and Synthia (Ros et al., 2016), three datasets providing high quality instance and semantic boundaries. Despite the inconsistent thickness of annotated boundaries, we also compare our method on BSDS500 Arbelaez et al. (2010). The dataset contains a training size of just 200 images as well as relatively low image resolution. For this evaluation, every method is trained on the BSDS500 training set, using the network pretrained on Mapillary Vistas. We compare VT against three different losses in binary representation; the dice loss (DL) (Dice, 1945), a weighted combination of Dice loss and cross-entropy (DCL) (Deng et al., 2018) and the weighted cross-entropy (WCL) (Xie & Tu, 2015). As part of the ablation, we further compare our method against the Distance Transform (DT) representation of boundaries which predicts a field of distances to the closest boundary. This is trained with a simple L1 loss between the ground truth $d_{gt}$ and the predicted distance $\hat{d}$. Each representation and training loss is computed using the same network architecture and optimization.

**Evaluation Metrics.** We evaluate each method based on the traditionally used R, P and F score in the ODS and OIS setups as well as on $asd_R$, $asd_P$, and $assd$ scores. For the traditional metrics for BSDS500 dataset evaluation (Arbelaez et al., 2010), we use an error tolerance of 0.0025 of the diagonal length of the image. This is lower than what is commonly used on the BSDS500 dataset, to account for the larger image sizes. To compute the surface distance metrics ($asd_R$, $asd_P$, and $assd$), each boundary representation is converted to a binary form with a thresholding operation. For the DL, DCL, WCL, and DT models, the threshold is fixed using a selected validation set for each dataset. For VT, instead, it is fixed for each test to the value of $-1$ of the divergence image, the same value added during inference before applying ReLU §4.2; points with lower divergence are classified as boundaries and the others as non-boundaries.

| Method | $asd_R$ | $asd_P$ | $assd$ | ODS | | | OIS | | |
|---|---|---|---|---|---|---|---|---|---|
| | | | | R | P | F | R | P | F |
| OP @$t_0$ | 5.37 | 3.29 | 4.33 | 0.814 | 0.878 | 0.845 | 0.819 | 0.874 | 0.846 |
| TP | 5.16 | 3.92 | 4.54 | 0.699 | 0.740 | 0.719 | 0.726 | 0.718 | 0.722 |
| TP @$t_0$ | 5.16 | 3.92 | 4.54 | 0.877 | 0.500 | 0.637 | / | / | / |
| TG | 6.02 | 3.26 | 4.64 | 0.621 | 0.850 | 0.718 | 0.621 | 0.850 | 0.718 |
| TG @$t_0$ | 6.02 | 3.26 | 4.64 | 0.464 | 0.917 | 0.616 | / | / | / |

Table 1: Comparison of the sensitivity to thickness in prediction and ground truth of the used metrics. OP is the original prediction, TP is the thickened version of the prediction and TG the thickened ground truth. In TP, we use the original ground truth and, in TG, the original prediction.

| Datasets and Method | $asd_R$ | $asd_P$ | $assd$ | ODS | | | OIS | | |
|---|---|---|---|---|---|---|---|---|---|
| | | | | R | P | F | R | P | F |
| **Cityscapes** | train: 2500 | | | validation: 475 | | | test: 500 | | |
| *VT* | 5.37 | **3.29** | **4.33** | **0.814** | **0.878** | **0.845** | **0.819** | **0.874** | **0.846** |
| DCL | **5.17** | 4.24 | 4.71 | 0.711 | 0.811 | 0.758 | 0.722 | 0.806 | 0.762 |
| DL | 5.40 | 6.51 | 5.96 | 0.758 | 0.747 | 0.752 | 0.747 | 0.760 | 0.754 |
| WCL | 6.42 | 5.98 | 6.20 | 0.773 | 0.756 | 0.764 | 0.755 | 0.779 | 0.767 |
| DT | 7.76 | 3.50 | 5.63 | 0.651 | 0.683 | 0.667 | 0.642 | 0.696 | 0.668 |
| **Synthia** | train: 6600 | | | validation: 800 | | | test: 1600 | | |
| *VT* | **1.73** | 1.61 | **1.67** | 0.767 | 0.877 | 0.819 | 0.767 | 0.878 | 0.819 |
| DCL | 3.60 | 2.69 | 3.15 | 0.682 | 0.754 | 0.717 | 0.710 | 0.730 | 0.720 |
| DL | 3.02 | **0.79** | 1.91 | 0.810 | 0.905 | 0.855 | 0.816 | 0.898 | 0.855 |
| WCL | 1.76 | 1.81 | 1.79 | **0.874** | **0.929** | **0.900** | **0.888** | **0.927** | **0.907** |
| DT | 4.72 | 2.76 | 3.74 | 0.786 | 0.840 | 0.812 | 0.782 | 0.846 | 0.813 |
| **Mapillary Vistas** | train: 17000 | | | validation: 1000 | | | test: 2000 | | |
| *VT* | 3.99 | **3.20** | **3.60** | 0.761 | **0.857** | **0.806** | 0.778 | **0.842** | **0.809** |
| DCL | 4.64 | 4.06 | 4.35 | 0.670 | 0.807 | 0.750 | 0.724 | 0.784 | 0.753 |
| DL | 5.16 | 3.28 | 4.22 | 0.735 | 0.787 | 0.760 | 0.733 | 0.792 | 0.761 |
| WCL | **2.86** | 5.67 | 4.27 | 0.759 | 0.730 | 0.744 | 0.767 | 0.763 | 0.765 |
| DT | 9.42 | 4.83 | 7.13 | **0.856** | 0.271 | 0.412 | **0.856** | 0.271 | 0.412 |
| **BSDS500** | train: 200 | | | validation: 100 | | | test: 200 | | |
| *VT* | 5.06 | 6.59 | 5.83 | **0.72** | 0.638 | 0.676 | **0.721** | 0.637 | 0.676 |
| DCL | 6.44 | 6.14 | 6.29 | 0.598 | 0.559 | 0.578 | 0.597 | 0.560 | 0.578 |
| DL | 7.99 | 5.81 | 6.90 | 0.540 | 0.534 | 0.537 | 0.543 | 0.531 | 0.537 |
| WCL | **4.04** | 7.28 | **5.66** | 0.660 | 0.718 | **0.688** | 0.662 | 0.716 | **0.688** |
| DT | 6.44 | **5.30** | 5.87 | 0.395 | **0.860** | 0.541 | 0.395 | **0.860** | 0.541 |
| Human | / | / | / | / | / | 0.8 | / | / | 0.8 |

Table 2: Evaluation results on the Cityscapes, Synthia, Mapillary Vistas and BSDS500 datasets. For each dataset we indicate the number of images respectively in the train, validation and test set.

## 5.1 METRIC ANALYSIS

We first show an analysis to support the use of the surface distance metrics (Yeghiazaryan & Voiculescu, 2018), $asd_R$, $asd_P$, and $assd$ over R, P, and F score in ODS and OIS conditions. We use the VT method results on the Cityscapes dataset, where boundaries have a constant 2-pixel thickness. To show the sensitivity of the metrics to different thicknesses in the prediction and ground truth, we compare the original prediction (OP) with the scores achieved when *doubling* the thickness of the prediction (TP) or of the ground truth (TG). When computing the R, P, and F score, we use two experimental setups: *(i)* changing the used prediction threshold in an ODS and OIS fashion and *(ii)* and keeping it fixed to the same value $t_0$ used in OP. For the surface distance metrics, the threshold is always fixed as in every other experiment.

From the results in Table 1, it is clear that the R, P and F scores are strongly dependent on the relative thickness of ground truth and prediction with variations in the F score of over 27% versus only a 7% change in $assd$. The small influence of thickness shows that the surface distance metrics are suited to evaluate predictions without post-processing. Furthermore, the evaluation also shows that the metric does not provide an advantage to the proposed method for its thin prediction.

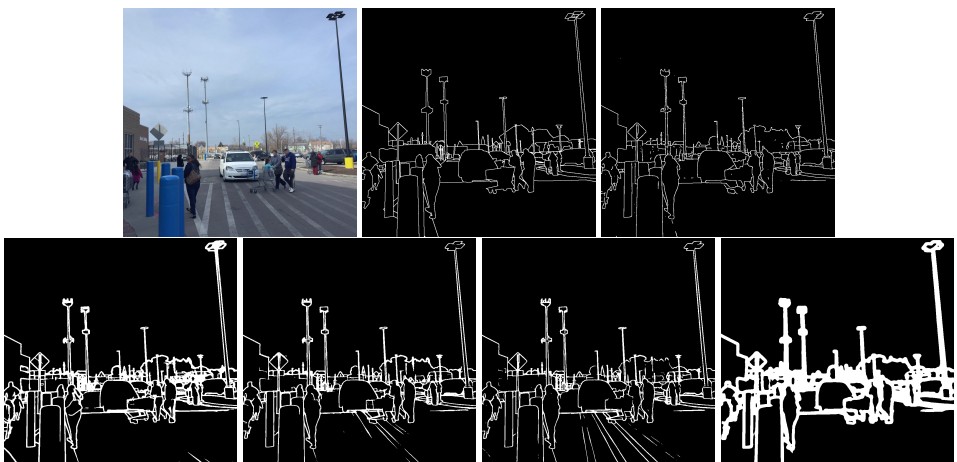

Figure 3: Qualitative comparison between methods on a Mapillary Vistas image from the test set. From left to right on the first row: the original image, the ground truth and the prediction using VT. On the second row: the prediction using WCL, DCL, DL, and DT.

## 5.2 REPRESENTATION PERFORMANCE COMPARISON

In this section, we show the performance of each representation on the four datasets considered. We do not apply non-maximum suppression so as to keep similar post-processing conditions throughout all methods and to evaluate the methods under the same conditions, so they could be integrated in other tasks §1. On Cityscapes, Synthia and Mapillary Vistas, our method consistently outperforms the other boundary representations in terms of $assd$, the metric less affected by the prediction thickness. Furthermore, it achieves competitive results for all other metrics, being the best performing method on Cityscapes and Mapillary Vistas in terms of F score. Throughout Table 2, it is possible to see that VT is the most stable method on the F score, being able to predict uniformly thin results. On BSDS500, VT is the second best performing method on $assd$ and F score, with a strong drop in overall performance given the dataset limitations. The VT field, in particular, suffers from the inconsistent annotations as they change the morphology of the field on and away from the boundaries, with errors that are not limited to the isolated pixel. Despite not achieving the best score on BSDS500, the result shows that VT is also able to predict low-level image contours without semantic meanings. To show its full capabilities in such task, it will be necessary to evaluate on a larger dataset with a clear definition of boundary, which is not currently available up to our knowledge.

From the qualitative comparison between predictions in Fig. 3, it is evident that the Vector Transform is the only method able to predict crisp boundaries without requiring any non-maximum suppression. A particularly important evaluation here is that of DT versus VT. We observe from the results that DT prediction in particular tends to detect thick boundaries, since a higher than 0 threshold is required to obtain enough recall, in turn leading to thick detection. The discussions provided at the end of §3 is directly relevant to its performance.

## 6 CONCLUSION

Despite being an old computer vision problem, boundary detection remains an active research field, both due to the many issues involved and its potential for other vision problems. In this paper, we propose the Vector Transform and theoretically demonstrate its equivalence with a surface representation of boundaries. The substantially different representation and corresponding approach of the Vector Transform automatically solve two of the biggest challenges: class imbalance in the prediction and the ability to output crisp boundaries. We show the high performance capabilities of the representation and a few related tasks that can be tackled from such an approach, with extensive evaluations. The formulation proposed in our work opens possibilities of new approaches on the use of boundaries and adds research questions on its applicability for several other related tasks.

**Acknowledgements.** This research was funded by Align Technology Switzerland GmbH (project AlignTech-ETH). Research was also funded by the EU Horizon 2020 grant agreement No. 820434.

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

## A    Vector Transform in Practice

There are some aspects of Vector Transform which requires special attention in several non-ideal cases. The first is the case of ground truth generation in datasets that were not designed with the interpretation of boundaries as surfaces. The second is the case of theoretical and practical aspects related to discontinuities and discretization.

### Ground Truth Generation

We now explain how in practice, we extract the Vector Transform from the ground truth boundary images to train the proposed method. For non-boundary pixels, the solution of the Vector Transform is clearly defined as the direction towards the closest boundary. In practice, to mitigate for discretization-related errors, the closest boundary pixel is computed averaging the multiple closest ones. This has the same effect of interpolating a planar surface on the set of closest boundary pixels and computing the normal direction to that surface. In the extreme case in which there are multiple equally distant boundaries, only one is chosen randomly as the closest one and the direction is computed towards it.

For boundary pixels, there can be multiple scenarios to take into account depending on the dataset quality:

- If the boundary is taken from a segmentation image, it can be treated as a non boundary pixel with its direction computed toward the closest set of pixel outside its semantic class/instance object.

- If the boundary is a manually annotated single pixel boundary, it is possible to consistently pair it to one of the neighboring non-boundary pixels and use its same vector direction. Any neighboring non-boundary pixel can be taken as long as the selection process remains consistent for each case.

- In case of thick boundaries, it is possible to devise a technique to assign a direction to each of its pixels - such as the direction of the closest non-boundary pixel and similarly to the previous point in undecided cases. However, this is not a proper definition of a surface boundary and, in practice, it is not considered in any of the tested cases.

### Discontinuities and Discretization

As explored in §3, the invertibility of the vector transform does not strictly satisfy at discontinuities. Around the infinitesimal neighborhood of the discontinuous boundary, the divergence is evaluated using an approximate normal. It can be observed that under a large number of circumstances, the transform can yield lower than -2 or -2 value in many discontinuities. However, the discretization and/or its combination with the discontinuities can impact the transform, where we may observe a higher than -2 divergence of the field. Fortunately, the representation provides a relatively large margin between the divergence of a non-boundary point from that of any other point. Such a margin provides the robustness required in order to predict correct boundaries even in crowded regions.

## B    Additional Results Analysis

In this section we report additional results with qualitative visualizations to support the observations in Section 5 and an additional experiment to analyze prediction profiles of DT versus VT measures around a predicted boundary. We show qualitative results obtained with every method on an image taken from each of the datasets. From the qualitative comparison of predictions in Fig. 4, it is evident that Vector Transform is the only method able to always predict crisp boundaries without requiring any non-maximum suppression. This provides a significant advantage, particularly in crowded regions where traditional methods require NMS post-processing to be used for downstream tasks. Therefore, our method provides a strong advantage when used at training time as an aide for different tasks.

Additionally, for the prediction profile experiment we measure the divergence of VT prediction along the normal direction of the boundary at an increasing distance. We compute the divergence versus distance for each predicted pixel. For these measurements at each distance we compute the mean and standard deviation. We perform the exact same experiment for the DT values instead of

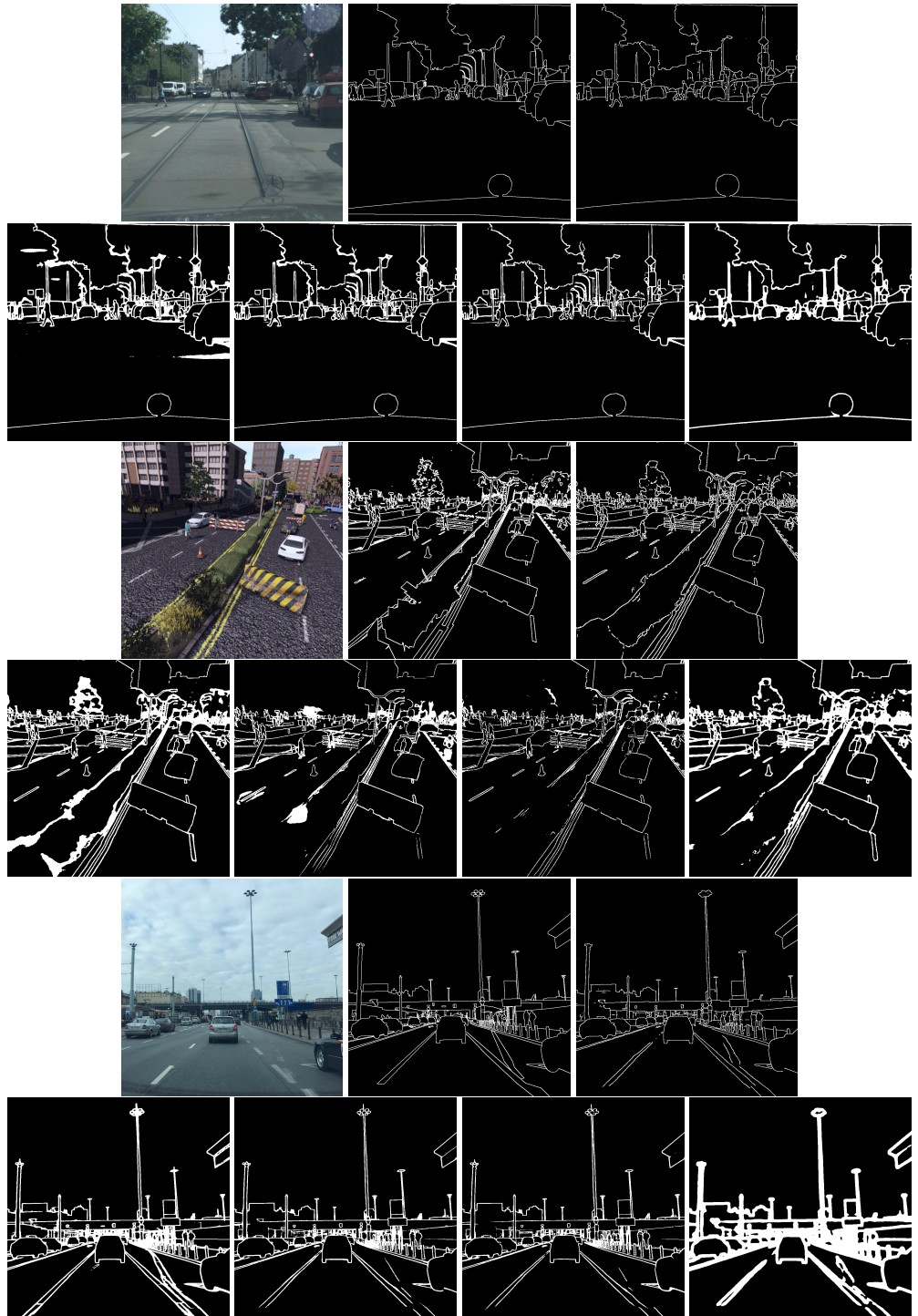

Figure 4: Qualitative comparison between methods. For each of the three examples, from left to right on the first row, there is the original image, the ground truth and the prediction using VT. On the second row, there is the prediction using WCL, DCL, and DT. From top to bottom, the first image is taken from Cityscapes test set, the second from Synthia and the third from Mapillary Vistas.

VT divergence. Both results are plotted in Fig. 5. The mean VT divergence (or DT value) versus the distance along the normal is plotted in black while the shaded region shows the standard deviation of the measure at each distance. Note how VT divergence value quickly saturates to around 0 from

-2 within a single pixel, while having an extremely low standard deviation. On the other hand, the DT prediction is mostly linear w.r.t distance by design but shows a large uncertainty around the 0 distance.

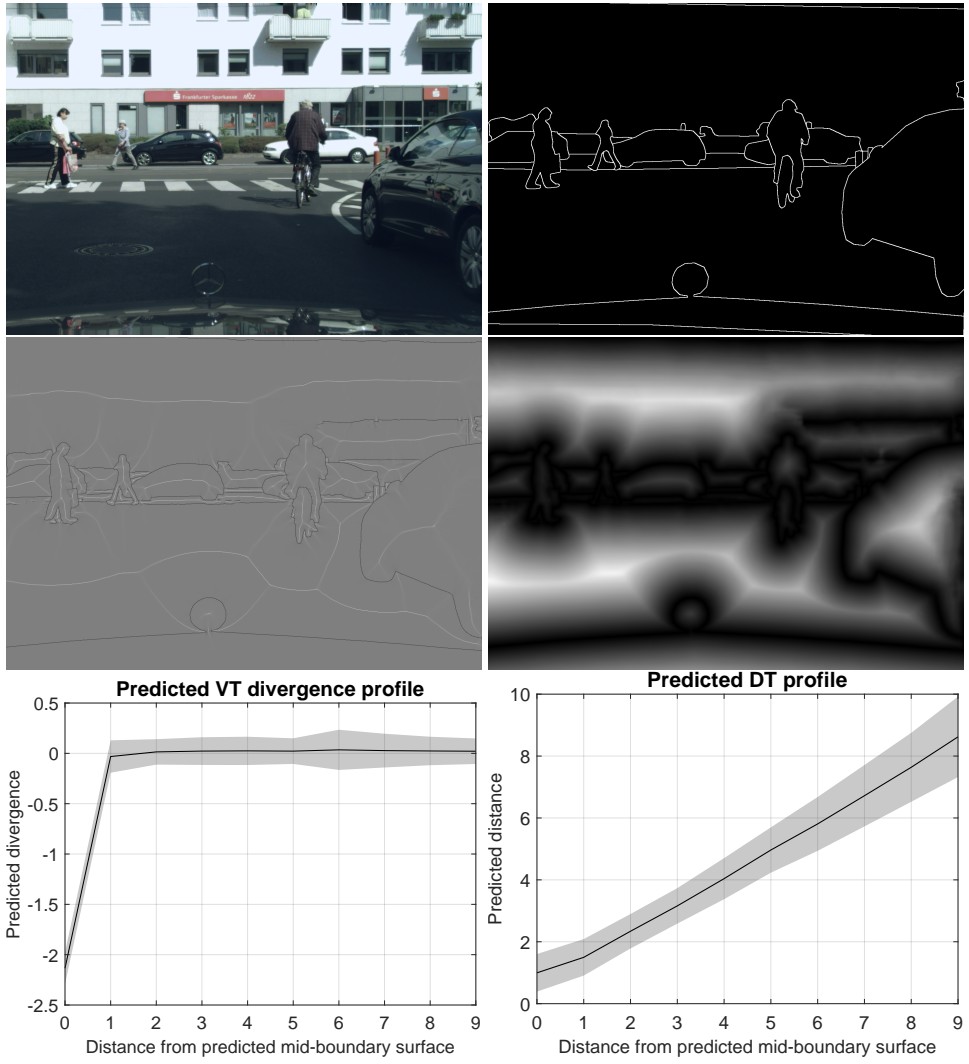

Figure 5: Comparison of the predictions using VT and DT on a randomly selected image from Cityscapes dataset. Going from top to the bottom row, we show the image (left) with the corresponding ground truth (right), the predicted divergence (left) and predicted DT (right) and the two prediction profiles. The profiles show the divergence on predicted VT (left) and the predicted DT (right) versus the distance from the mid-boundary surface for the VT and DT methods. The plots show the mean (black line) and the standard deviation (gray shading) around it.

## C    BOUNDARY DIRECTION ESTIMATION

As our method outputs the VT for each pixel, when the boundaries are represented inside the image, it is natural to extract the boundary direction. Differently from other methods (Maninis et al., 2017), no additional module or post-processing is required to estimate the direction. Furthermore, our method is able to predict continuous angles without the necessity to select from a discrete set of values.

In Fig. 6, we show a qualitative representation of our boundary direction estimation on three example images from Mapillary Vistas to have a visualization of the prediction quality. From a quantitative

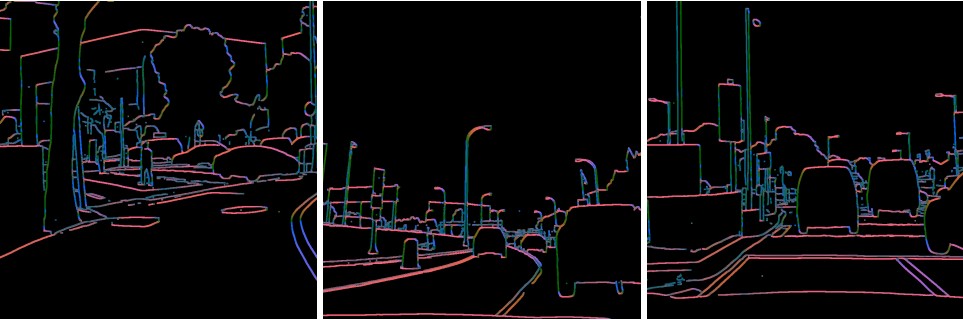

Figure 6: Qualitative results of the boundary direction estimation on three images of Mapillary Vistas test set. In the figures, the direction of the boundaries is plotted to the boundary pixels, which are thickened to ease visualization. Vertical lines correspond to colors in the range of green and blue that gradually turn into red and pink for horizontal lines.

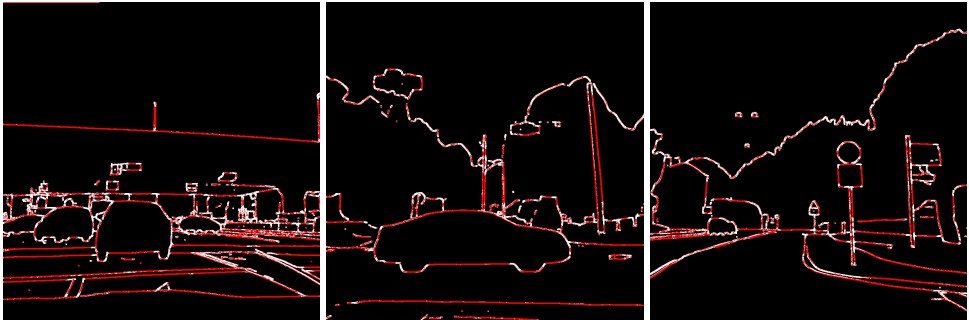

Figure 7: Qualitative example of line detection using the VT field on three images of Mapillary Vistas test set. In the images, the straight lines in the boundaries are identified with the red colors. For visualization purposes, the boundaries have first been artificially thickened before detecting straight lines.

standpoint, the predicting VT field can achieve a root mean squared error on the estimated angle $\theta$ of 7.2 degrees.

## D  STRAIGHT LINE PROPOSALS

Given that in our method each boundary pixel has a direction feature, it is natural to try and solve the task of straight line proposal generation for detection without any specific supervision. A straight line is considered a boundary for which the direction does not change in neighboring pixels. To detect such points, it is possible to apply a simple algorithm:

- First the VT field is converted using only the absolute value of the two channels. In this way, differences of vectors having same orientation but opposite direction on the two sides of a boundary are removed.
- Then the derivative of the two channels are approximated using a Sobel filter and their absolute values are summed in a pixelwise manner.
- In the obtained image, pixels with a high value indicate places where the orientation changes while the low values show constant orientation. Therefore, we define a threshold ($t = 0.05$) and consider part of a straight line the boundary pixels with a value below the threshold.

In Fig. 7, we show some qualitative results obtained with the above method. It is possible to identify short straight lines detected in areas of the image without an apparent straight line. This is due to the small dimension ($3 \times 3$) of the derivative kernel used to detect a straight line and could be solved by filtering the prediction. We show the result without postprocessing as a proof that our method can be applied no matter what type of line detection is needed, from small segments to long lines.

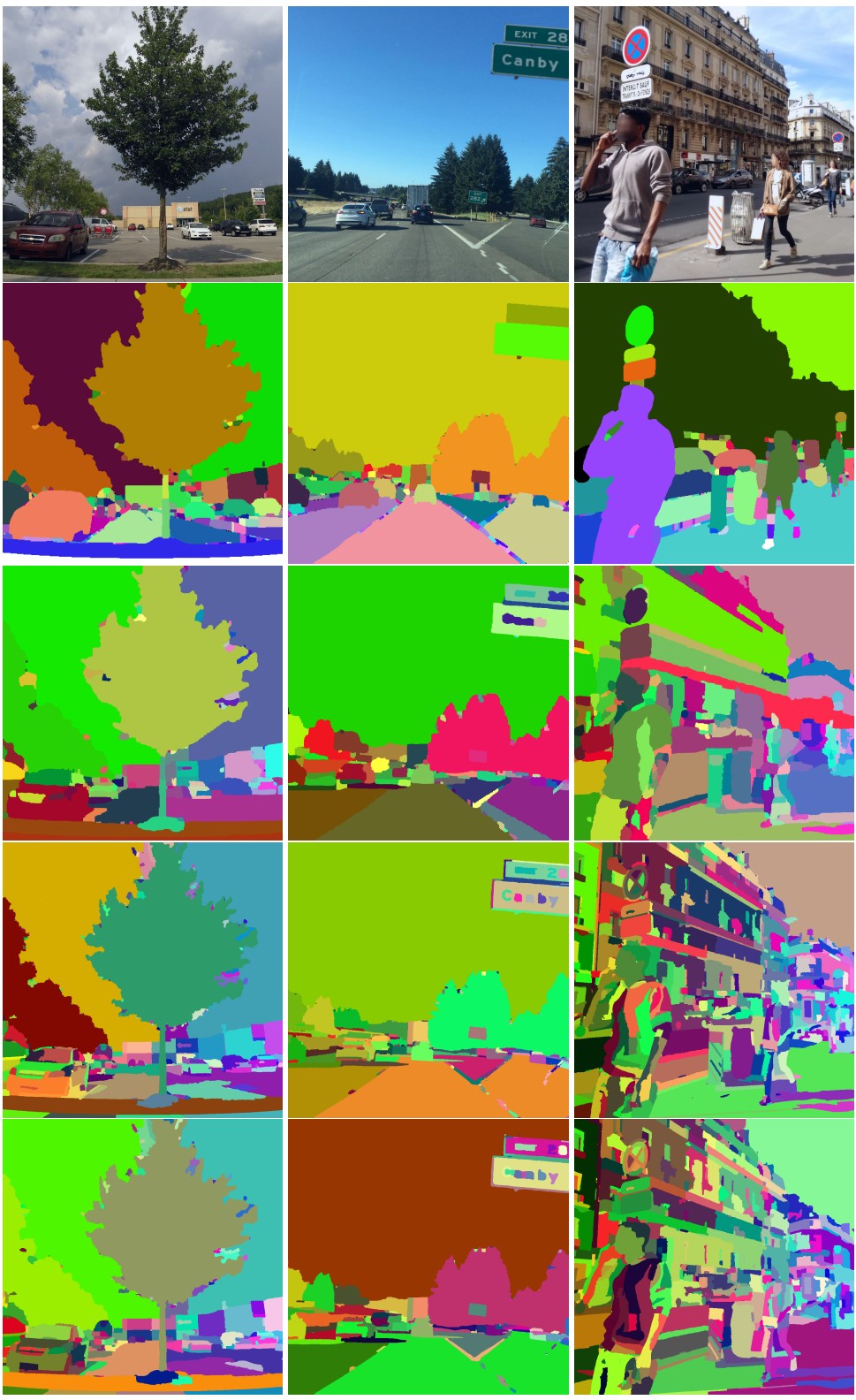

Figure 8: Examples of superpixels obtained on three images of Mapillary Vistas test set using four different methods. From top to bottom: the original images, the superpixels obtained using the VT field, COB (Maninis et al., 2017), MCG (Arbeláez et al., 2014), and SCG (Arbeláez et al., 2014), respectively.

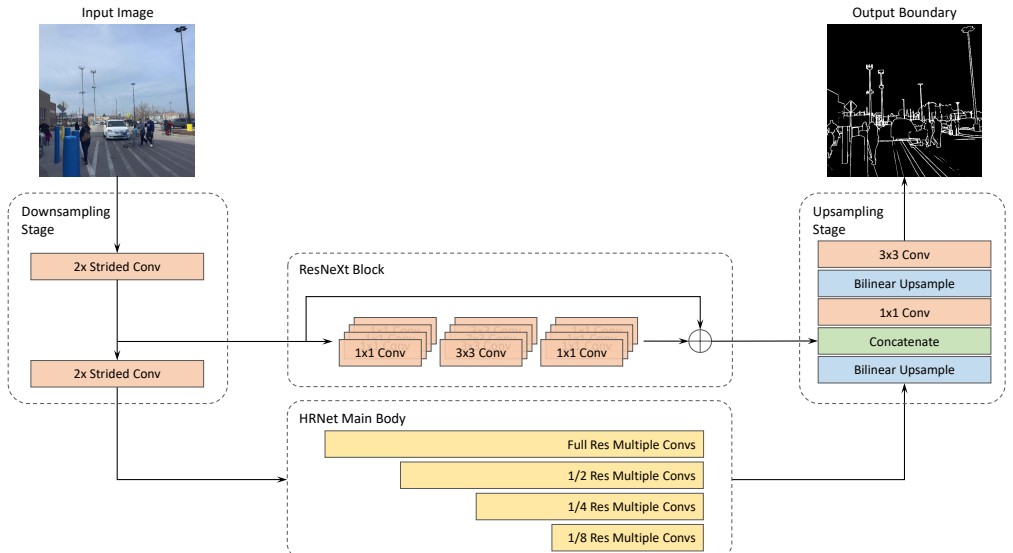

Figure 9: **Network architecture overview**. Schematics of the network architecture highlighting the way HRNet is used and how full resolution boundaries are predicted. Each convolution, except from the last one, includes batch normalization (Ioffe & Szegedy, 2015) and a ReLU activation (Nair & Hinton, 2010). As output, we show the predicted boundary; this can be obtained from any method as we use the same network changing only the post-processing.

## E    SUPERPIXEL

The VT field can also be used to create superpixels without any specific supervision. More specifically, when they are trained on semantically meaningful boundaries, they can be used to extract objects or object parts. This can be done without obtaining the partial result of boundaries, using only the fields and applying region growing algorithm on it. Specifically, we use the following algorithm:

- First, divergence is computed on the VT field using a Sobel filter as done to obtain boundary pixels. The high divergence values are the source points in the field and are treated as centroids of the parts. In case there is a connected part with high divergence, it is considered to be a single centroid region.

- Each pixel is associated to a two dimensional point based on its coordinates on the image grid. The point is then moved (its coordinates are changed) following the opposite of the field direction, towards the center and away from the border. This continues until the algorithm converges or for a maximum number of steps. This results in each pixel being associated with a point that has been moved to a different position from the original.

- Based on the position of the obtained point, the pixels are then assigned to the closest part centroid. Thanks to the complex shapes that the centroid regions can assume, the resulting clusters can have high structural complexity and represent complete objects or large parts. In the special case of pixels with an associated point that falls outside of the image border, an assignment to a part centroid cannot be done. These pixels are then clustered using DBSCAN (Ester et al., 1996) algorithm using as features the position of the associated points.

The technique benefits from the possibility of being made highly parallel with the clustering algorithm that only needs to be applied on a limited number of well separated points. We show some results obtained by grouping pixels based on the predicted Vector Transform in Fig. 8. We compare to some well-know superpixel methods, i.e., Convolutional Oriented Boundaries (COB) (Maninis et al., 2017), Multiscale Combinatorial Grouping (MCG) (Arbeláez et al., 2014), and Singlescale Combinatorial Grouping (SCG) (Arbeláez et al., 2014), whose results are generated by thresholding the occlusion boundaries (Ultrametric Contour Map, UCM (Arbelaez et al., 2010)) using the optimal threshold of each method. It is visible that the VT field can group an object with high complexity

as a single superpixel, which can ease downstream tasks that benefit from such a characteristic. In contrast, previous methods tend to create a higher number of clusters whose boundaries are not always coincidental with the complete objects. This task further justifies our re-definition of boundary detection.

## F  NETWORK ARCHITECTURE AND TRAINING

In this section, we provide more details on the network architecture and the training procedures. Figure 9 shows a schematic of the architecture used. The architecture is chosen to take into account both high and low resolution details as commonly done by boundary detection methods (Liu et al., 2017; Xie & Tu, 2015). More specifically, the use of a ResNeXt (Xie et al., 2017) block to process the high resolution signals and the upsampling stage are inspired by Deng et al. (2018). Analyzing the architecture in details, we can identify four different sections:

- **Downsampling phase**: the image goes through two strided convolution layers which reduce the image resolution and increase the number of channels to $64$. This is part of HRNet (Wang et al., 2020) but is represented separately for a better understanding.

- **HRNet Main Body**: this is the main part of HRNet (Wang et al., 2020) which extracts multi-resolution features subsequently after the two strided convolutions. The input and output of this block have the same resolution.

- **ResNeXt Block**: this is a single ResNeXt (Xie et al., 2017) block that is used to extract complex features to be used while retrieving the full resolution. More specifically, the block has cardinality 4, using the same notation as Xie et al. (2017).

- **Upsampling phase**: the final part of the network takes as input, the output of HRNet and up-samples it to formulate full resolution predictions. First, the input is bilinearly upsampled and the result is merged to the output of the ResNeXt block concatenating the two and applying a pixel-wise fully-connected layer. Finally, the result is further bilinearly upsampled and one final convolution layer is used to predict the output representation.

The training protocol used for the network resembles the one used in Panoptic-DeepLab (Cheng et al., 2020). More specifically, we apply an initial learning rate of $0.001$, the 'poly' learning rate scheduler (Liu et al., 2015) and a batch size of $32$. The images are augmented using random resizing, random horizontal flipping and randomly cropping the resulting image to the size of $512 \times 512$ irrespective of the dataset. The dimensions for randomly resizing the shortest side are from the set of dimensions $\{512, 640, 704, 832, 896, 1024, 1152, 1216, 1344, 1408, 1536, 1664, 1728, 1856, 1920, 2048\}$. Inference, instead, is done at a single image resolution irrespective of the method applied. For Cityscapes (Cordts et al., 2016) and Mapillary Vistas (Neuhold et al., 2017) the shortest edge has dimension $1024$ pixels and for Synthia (Ros et al., 2016) it is 960 pixels.

