# OpenReview forum: "Zero Pixel Directional Boundary by Vector Transform"
_ICLR.cc/2022/Conference — ICLR 2022 Poster_

### Official Review · Reviewer_5b3T · 2021-11-02

**Correctness:** 3
**Technical Novelty And Significance:** 3
**Empirical Novelty And Significance:** 3
**Recommendation:** 6
**Confidence:** 3

**Main Review:**

+ novel representation of boundaries inspired by classical implicit functions and combining it to a modern NN

+ achieves thin boundaries which can be useful in many applications

- The neural network architecture and training details are not very clear. The paper mentions HRNet is used with some skip connections and conv layers added, but in my opinion it is very difficult to reproduce the authors' experiments given the current manuscript.

- Only one backbone architecture is tested for all losses, and Vector Transform only outperforms others in some but not all metrics as shown in Tables 3 and 4. It is not convincing that the vector transform representation can be useful and outperform the binary representation in practice

**Summary Of The Paper:**

This paper proposes treating boundaries as 1D surfaces using vector transformations in Equation (1).
The paper then proves a 1-to-1 relationship between the vector field representation and a binary boundary map.
The paper further learns the vector representation into an NN by using a variant of HR-Net that outputs a dense two-channel image with x and y component of the vector field and train it with Mean Sqaured Error loss.
Since the predicted boundaries using this method can be infinitely thin, so the paper proposes using average surface distances to evalute the performance, and shows improved results of their method comparing against three other losses using the same NN architecture.

**Summary Of The Review:**

The paper takes a first-principles approach to thin boundary estimation. The novelty outweighs the concerns mentioned. The paper is worth known to the community.

---

> ### Author Response · Authors · 2021-11-15
> **Response to reviewer 5b3T**
>
> We thank the reviewer for their time and effort in reviewing the paper. We appreciate that the reviewer valued the first principles approach to the problem and acknowledged our theoretical contributions as well. We apologize for a few typos and errors that have been corrected in the current version.
>
> - *Hard to reproduce*: We understand the difficulty in perfectly understanding the architecture as, despite trying to precisely explain the architecture used (Section 4.1), it is often difficult to reproduce that with minimal effort.
> To improve on the reproducibility side, we plan to release the code when the paper is published and we will also add an appendix section to better explain the architecture and the training procedure with the help of a figure. This will be available in the next draft of the paper before the end of the rebuttal period.
> - Assessing the doubts about the actual advantage of the representation, we point out that VT shows to perform overall better than any other single method in the current experimental section. It is always the best method in assd and the best method in terms of ODS and OIS F score in two out of three datasets (these metrics evaluate the overall quality better than precision and recall taken alone). Furthermore, it is only the first work that uses VT representation for boundaries, additional avenues exist in order to further improve the accuracy such as an optimized architecture for VT and its implicit attention modeling.
> Considering the architecture, HRNet was chosen as it combines both high- and low-level features (as explained in Section 4.1), which has been shown to be important to tackle boundary detection by binary representation methods (HED).
> Furthermore, as a sign that the representation performance is not limited to the use of a single architecture, we report here some results obtained on the assd metrics while developing the method using a Deeplab architecture on Cityscapes:
>   - VT: 4.22
>   - DCL:    4.95
>   - DL:    6.51
>   - WCL:    7.35
>   - DT:    5.66
>
>   As it can be seen, in this limited testing scenario, the gap between VT and the other methods is higher. Despite the good results, it was not selected for extensive testing as HRNet combines high- and low-level details at more levels, as suggested for binary boundary detection (HED).
> Other advantages of the VT representation include providing sharp boundaries in a differentiable manner (Section 4.2), the absence of hyperparameters to tune (Section 1, Section 4.1, 4.2), and the possibility to apply it to other tasks with minimal overhead. We have shown some examples of such tasks in the appendices (Section 4.2 last paragraph, Appendix C, D, and E).

---

### Official Review · Reviewer_GvzB · 2021-11-08

**Correctness:** 3
**Technical Novelty And Significance:** 3
**Empirical Novelty And Significance:** 2
**Recommendation:** 6
**Confidence:** 4

**Main Review:**

The writing is clear and easy to follow. The whole idea is simple and the result looks quite promising. The experiments seem sufficient.

I have some concerns regarding the details:

1 The authors proposed a "simpler" solution in inference. However, the implementation confused me somehow: The authors defined \tilde{I} in eq. 6 as a support image. But I did not see how $\tilde{I}$ was involved in the following computation. $\tilde{I}$ disappeared in eq. 7 and 8. What is the input image to the network? $I$ or $\tilde{I}$?
I did not see the benefit of using this new approach to obtain the boundary image either. In fact, as the authors mentioned, we can simply use a 2x2 kernel, which is also very simple and clear.

2 The authors discussed the drawbacks of the ODS and OIS metrics and proposed using asd series metrics in addiction. However, the authors did not mention the potential drawback of asd metrics: I think asd is very sensitive to prediction noise, especially in asd_P. Think about the predicted boundary map containing an outlier that is far from any groundtruth edge point. This simple noise point can contribute a lot to the final asd_P score.

3 Using Dice Loss as a baseline is fine. The authors might also want to include more recent development on dice loss, for example, clDice loss:

Shit, Suprosanna, Johannes C. Paetzold, Anjany Sekuboyina, Ivan Ezhov, Alexander Unger, Andrey Zhylka, Josien PW Pluim, Ulrich Bauer, and Bjoern H. Menze. "clDice-a Novel Topology-Preserving Loss Function for Tubular Structure Segmentation." CVPR. 2021.

The above method also claims the robustness against data imbalance and efficacy on topology preserving.

4 I am a little surprised to see that DT performs much worse than the proposed VT method. I think they are pretty similar. I wonder how the authors determined the threshold for DT, which will determine the thickness of the edge? Why did the authors use L2 loss in VT while L1 in DT? Is L1 better for DT?

5 I appreciate that the authors experimented on the BSDS500 dataset, a widely used dataset for edge detection evaluation. Though the proposed method did not achieve the top performance on this dataset, it would be great if the authors could have more insight into why WCL performs relatively better in this scenario. I also suggest moving this experiment from the supplementary material to the main script.

**Summary Of The Paper:**

This paper is about learning-based edge detection. The authors proposed a new loss function for end-to-end edge detection to overcome the label imbalance and edge thickness problems. The loss function is based on the vector transform field, which is closely related to distance transform. The final edge map can be easily obtained by thresholding the divergence of the predicted vector transform field. The authors compared the proposed VT loss with the dice loss, the combination of dice loss and cross entropy loss, weighted cross entropy, and the distance transform based loss, on Cityscapes, Mapillary Vistas, Synthia, and BSDS500 datasets. The experiment results show the promising performance of the proposed VT loss on the edge detection task.

**Summary Of The Review:**

Overall I think this is an OK paper. The VT representation of an edge map seems novel to me. I like the simple idea which works pretty well on several datasets. The implementation is straightforward without bells and whistles. However, I still have some concerns about the details including the support image, DT performance, etc.

---

> ### Author Response · Authors · 2021-11-15
> **Response to reviewer GvzB (part 1)**
>
> We thank the reviewer for their time and effort in reviewing the paper. We appreciate that the reviewer valued the simplicity and novelty of the idea behind the VT method, as well as its performance on several datasets.
>
> We try now to address each of the concerns pointed out:
>
> 1. We thank the reviewer for pointing out the clarity problems in Section 4.2. We wanted to highlight the method using the support image, as it is an easy technique to localize zero-pixel boundaries. 2x2 convolutions, instead, localize the prediction in a pixel position, and obtaining zero pixel boundaries requires some post-processing. All mentions of images in that context should imply the vector field image and equations (7) and (8) should be written with $\tilde{\mathsf{v}}$. We have corrected this part by defining a mathematical operator for equation (6), while emphasizing that this is needed only in case a 0-pixel thick boundary is desired.
> 2. Regarding the metrics, we agree that the high sensitivity to missed and false predictions can be a drawback, depending on the application. However, ODS and OIS metrics penalize each pixel without a 1-to-1 match in the same way, no matter how close to a real prediction it is. Instead, $asd_P$ and $asd_R$ are strongly penalized only when there is an actual missed prediction or a fake prediction completely off an actual boundary. Therefore, we still believe that the surface distance metrics are still extremely valuable to provide a good evaluation of the prediction quality. Nonetheless, we have updated Section 4.3 to highlight this limitation.
> 3. Thanks for pointing out this additional reference method. We will cite and discuss the method. Despite reporting good performances in terms of prediction crispness and topology preservation, it has strongly different properties compared to VT (which provides strictly thin boundaries and absence of class imbalance).
> The method, designed for more general segmentation tasks, relies on having a loss that extracts the skeleton of the ground truth as well as the prediction. For thin boundaries, this operation has no effect on the ground truth as it is the skeleton version of itself. This leads to a low Tprec (Equation 1) score when compared to cases with thick ground truths. The sensitivity term (Tsens in Equation 1), instead, favors thick prediction over thin ones as that increases the overlapping (as long as they don’t change the topology). Therefore, the method does not encourage predictions to be crisp nor adds further class imbalance robustness compared to the Dice loss alone.
> Nonetheless, given its good demonstrated performance, we are currently running experiments with this loss as well and we expect partial results to be available in the following week. Unfortunately, we cannot assure the results on every dataset will be available before the end of the rebuttal period, but we will report them once we have all the experiment results.

---

> > ### Author Response · Authors · 2021-11-15
> > **Response to reviewer GvzB (part 2)**
> >
> > 4. We appreciate the reviewer’s concern about the large gap in performance between DT and VT, two representations that are strongly related by the Jacobian. We understand that more clarity and discussions will be appreciated also in the main paper. Discussions on the training sensitivity of VT vs. DT and SDF were provided in Section 3, Paragraph “Vector field and the Distance Field”. First, despite the strong relation between DT and VT, without the findings in Section 3, it is not obvious that VT can be used for surface representation and inference. We refer the reviewer to Property 3.2 and its proof. Equations (2) and (3) then further hints at how the VT contrast is different from the DT contrast at the boundary. One can call it an inductive bias of the VT problem definition for encoding sharp boundaries. Please refer to the contrast of VT as described in the paragraph below equation (8) in the main text. Under prediction noise, DT has poor localizability, meaning that it is very difficult to find the zero-activation threshold at inference. Partly for that reason, 3D implicit functions use an SDF representation instead, where zero-crossings are easier to detect. Please look into the main paper tables and figure 4 in Appendix B where DT has more missed predictions despite having the thickest boundaries. This clearly cannot be solved by choosing better thresholds. On the other hand, VT has excellent localizability under the prediction noise at inference, due to the sharp contrast of divergence. We further support this claim with figure 5 (Appendix B) which shows the different profiles of the predicted VT divergence and the predicted DT when moving away from the predicted mid-boundary surface. The large uncertainty in the predicted DT value inside an image doesn’t allow to set a threshold able to define crisp and complete boundaries.
> > Another point to note is that DT, despite reducing the class imbalance compared to binary boundary prediction, has an implicit bias. As one can define an order in its values, its training will favor values around the weighted average of its typical range. On the contrary, VT doesn’t suffer from such a drawback. We have modified the last paragraph in Section 3 to include these theoretical reasonings for the suitability of VT over DT.
> > Answering the question about the selected threshold, we tried 20 values uniformly taken in the [0.5, 4] range. (the number of points used is the same for every method except VT where the value was set prior to any major experiment, using the theoretical reasoning of equation (3). (Section 4.1, 4.2))
> > Regarding the used loss, given that DT is a scalar field, there is actually no difference in applying to it an L1 or L2 loss.
> > 5. Regarding the comment on BSDS500, we moved it to the main body of the paper. Regarding the performance of WCL, a reason why it outperforms VT is that the dataset boundary annotations are ambiguous leading to a variable thickness in the ground truth. VT is not designed to handle such annotations and therefore gives a lower accuracy. Owing to the surface definition of boundary as VT, this is the single most important limitation of the proposed method which restricts its usage in older datasets with inconsistent annotations. (Section 5 and 5.2)

---

> > > ### Author Response · Authors · 2021-11-27
> > > **Experiment on clDice**
> > >
> > > As a follow-up to the previous answer, we tested clDice method (with implementation corrected for bugs in the official repository) on Cityscapes with the following results:
> > >
> > > | method | $asd_R$ | $asd_P$ | $assd$ | $ODS_R$ | $ODS_P$ | $ODS_F$ | $OIS_R$ | $OIS_P$ | $OIS_F$ |
> > > | ------- | -------- | ------- | -------- | ------- | -------- | ------- | -------- | ------- | -------- |
> > > | clDice | 5.07 | 4.06 | 4.56 | 0.758 | 0.840 | 0.797 | 0.748 | 0.857 | 0.799 |

---

> > > > ### Comment · Reviewer_GvzB · 2021-11-30
> > > > **Responses to the authors**
> > > >
> > > > I thank the authors for the detailed the response, the extra experiments, and the revised draft. I am satisfied with the response and most of my concerns have been resolved.
> > > >
> > > > Overall, VT seems an interesting and promising representation for boundary detection. Though similar to DT, VT outperforms DT by a large margin and is more robust against noise.
> > > >
> > > > I appreciate that the authors discussed the limitations and drawbacks of the evaluation metrics and analyzed the possible reasons that VT did not perform the best on the BSDS500 dataset.
> > > >
> > > > So overall, I think this is a good paper and it could benefit a broader computer vision community.

---

### Official Review · Reviewer_CywP · 2021-11-08

**Correctness:** 4
**Technical Novelty And Significance:** 4
**Empirical Novelty And Significance:** 4
**Recommendation:** 8
**Confidence:** 3

**Details Of Ethics Concerns:**

No concern for ethical problems.

**Main Review:**

Strengths
1. Novelty
The paper proposes a novel idea of the surface representation of boundaries

2. Theoretical justification/discussions
The paper shows theoretical justification and discussions.

3. Experiments
Experimental results support the effectiveness of the proposed method

Weakness
1. The paper does not show the limitation of the proposed method.


**Summary Of The Paper:**

The paper proposes a method to detect boundaries in an image by interpreting boundaries as 1-D surfaces and formulating a one-to-one vector transformation function. The paper provides theoretical justification of the vector transformation representation. Experimental results show the effectiveness on publicly available datasets.

**Summary Of The Review:**

The paper proposes a new method with careful theoretical justification. Experimental results support the effectiveness. The paper is well-organized. The paper is beneficial to the readers.

---

> ### Author Response · Authors · 2021-11-15
> **Response to reviewer CywP**
>
> We thank the reviewer for the time and effort in reviewing the paper. We appreciate that the reviewer valued the careful theoretical justifications/discussions of the novel representations as well as the experimental results showing the effectiveness.
>
> *Limitation of the method*: The main limitation of our method is the necessity of a clear definition of boundary in dataset annotations (Appendix C previous version, Section 5 and 5.2 in the updated version). Each “wrong” boundary training annotation changes the topology of a large region around it and therefore substantially harms the learning capabilities. We could not use datasets with many undefined regions as they create boundaries with no real meaning. Similarly, the ambiguous definitions of boundaries in BSDS500 harmed the performance. However, this is not a problem if the datasets are appropriately labeled with consistent small thickness, as in Mapillary Vistas or Cityscapes (Section 5).

---

### Decision · Program_Chairs · 2022-01-20

**Decision:**

Accept (Poster)

**Comment:**

The authors proposed a new loss function for end-to-end edge detection to overcome the label imbalance and edge thickness problems. Overall, the proposed VT appears to be a useful representation for boundary detection. Though similar to DT, VT outperforms DT by a large margin and is more robust to noise. One reviewer recommends acceptance, two others recommend marginal acceptance. The main issues have been adequately addressed in the rebuttal.